# Dance to Prosper: Benefits of Chinese Square Dance in QOL and the Moderating Roles of Aging Stereotypes

**DOI:** 10.3390/ijerph192416477

**Published:** 2022-12-08

**Authors:** Honghao Zhang, Huiyuan Jia, Xin Zhang

**Affiliations:** 1School of Psychological and Cognitive Sciences, Peking University, Beijing 100871, China; 2College of Business Administration, Capital University of Economics and Business, Beijing 100070, China

**Keywords:** dance duration, attitude toward own aging, quality of life, midlife people

## Abstract

Background: By examining the effect of Chinese square dance duration through a positive activity model and discussing the impact of aging stereotype internalization, this study examined the relationships between dance duration, attitudes toward own aging (ATOA), aging stereotypes, and quality of life. Methods: 403 Chinese square dance participants were recruited to complete a 7-day diary survey in a cross-sectional design. Participants reported on their ATOA, aging stereotypes, perceived quality of life, and everyday dance participation during the week. Data were analyzed using latent variable structural equation modeling. Results: Increased dance participation improved quality of life, and the mediation by ATOA was determined. Positive and negative aging stereotypes separately moderated the mediating process. In general, people who had relatively stronger negative stereotypes benefited more from dancing duration, while people with stronger positive stereotypes felt no such dose effect. Conclusions: The results suggest that an attitudinal pathway explains the positive activity dose effect, and people with more negative aging stereotypes are encouraged to practice Chinese square dance to benefit from certain effects.

According to Chinese sports and fitness enterprises, there are approximately 80–100 million Chinese square dance lovers around the country, and the number is still increasing. In 2016, 1000 women gathered to perform Chinese square dance on World Fitness Day [1], and nearly 100,000 people spontaneously participated in the National Square Aerobics and Dance Games in 2018. These large gatherings suggest a fever for Chinese square dance. 

Though dancing is popular across all ages, it is worth noting that Chinese square dance seems to be a particular interest of *dama*s (middle-aged women). In fact, a recent national survey of more than 30,000 dancers reported that 88% of dancers were female, meanwhile 62% of the participants were aged from 35–65 [2]. Square dance seems to be a favorite of the middle-aged. 

Certain reasons may lie behind this phenomenon. In midlife, people begin to form personal views of their own aging experiences [3], and attitudes to aging formed in midlife can assume a critical psychosocial adjustment in old age [4]. In other words, in midlife people start to have awareness of aging and try to face it. Being involved in activities such as square dance might be one of their strategies. 

## 1. Chinese Square Dance: A Positive and Emotionally Meaningful Activity

A positive activity model [5] proposes the relationship between Chinese square dance and midlife well-being. Positive activity refers to simple, intentional, and regular practices that increase healthy thoughts and behaviors. In this model, the performance of positive activities produces positive emotions, positive thoughts, positive behaviors, and need satisfaction, which increase well-being. Certain positive activities are proved to cause positive consequences, and they can take many forms. Examples show that people who use their strengths in new ways [6] or affirm their values [7] report being more satisfied about their daily life. That is to say, engaging in positive activities leads people to construe life events more positively. 

In addition, positive bias in older adults [8] results in a cognitive knowledge of people’s preferences for how they use their free time. People will be guided to selectively attend to and retain positive information in their lives as they become older by a later established positivity effect [9]. This propensity for positivity also aids persons in midlife in seeing the positive aspects of their surroundings. Taken together, these factors suggest that as people age, they may be more open to participating in positive activities and discovering positivity in their experiences.

Another widespread theory, named socioemotional selectivity theory [10] adds insight to the existing model. The theory proposes that life span change in the preference of social interaction plays an important role in influencing peoples’ motivational goals and decisions. Meanwhile, SST theory points out that as their age increases, people prioritize emotionally meaningful goals more and are motivated to implement their social preferences [10]. Neuroimaging evidence also indicates that messages with a focus on social impact work more effectively for an older population [11]. 

Together, SST and the positive activity model better explain why middle-aged and older Chinese people choose to enrich their recreational lives with dancing. Firstly, square dance is a simple, repeated practice that people automatically choose to spend their time on because they gain positivity from it. It generates enjoyable and happy feelings, and satisfies social needs, as postulated by need theory [12,13]. Besides, studies have proved that Chinese square dance enlarges participants’ social circle and helps them to defeat loneliness [14,15]. Apparently, Chinese square dance helps people to fulfil social needs, such as providing a sense of belonging, security, and self-recognition [14,16]. In a collective context, it also helps to defeat depression and anxiety [17,18]. A study of middle-age women illustrated that in midlife, people are more likely to be motivated to stick with an activity that fits their social needs [19], and in China, square dance creates an environment for such needs. When people put more weight on psychological values such as social relations and mood, they are more likely for those reasons to participate in such activities [20] and be happy with them. In other words, by intentionally increasing square dance participation, people are precisely choosing to act pro-socially to gain greater happiness, and to spend their time in emotionally meaningful ways.

## 2. Dance Influences Quality of Life (QoL) through Activity Type and Duration

Quality of life (QoL) is a multidimensional, multilevel concept that measures “individuals’ perception of their position in life” [21]. The global QoL term includes personal reports of life satisfaction and subjective feelings of well-being, often defined in domains such as physical, psychological, social, and environmental evaluations of life [22]. Studies on QoL have a long history in examining the role of daily activity duration as a benefit of and motivator for it. According to activity theory, increased activity engagement [23] and greater frequency of participation in activities are related to higher levels of subjective well-being [24], which is an essential subjective indicator of QoL [25,26]. Many types of activities are believed to have such a dose effect. For example, spending longer time in leisure [27], social [28,29], and physical activities [20,30,31] have a positive association with QoL. 

Studies of leisure activities have mostly grouped activities according to whether they are predominantly mental, physical, or social, but activities such as dancing simultaneously embody more than one of these aspects [32]. Hence, a multidimensional profile is developed according to this consideration, and daily leisure activities with various combinations of mental, social, and physical involvement are grouped in a novel coding system to assess activities [33]. Authors have raised examples like golf, bowling, and dancing, which involve multiple needs of people. Dance is regarded as fulfilling the most needs including social interaction, attention, and balance needs [34]. When considering the combined benefit, such activities covering more than one of the mental, physical, and social components have a larger positive effect on health outcome performance in adults than those with fewer components [35]. Therefore, we believed that Chinese square dance will satisfy multiple needs as described by QoL and thus is a cogent indicator of people’s perceived QoL. More specifically, the more time spent on Chinese square dance, the more positive is the QoL perception expected among the dancers. Taken together, Hypothesis 1 assumes that the duration of Chinese square dance is positively related to perceived QoL. 

## 3. Mediation by Attitude toward Own Aging (ATOA)

Individuals pay more attention to their physical health and longevity as multiple challenges occur in midlife. Under such circumstances, positive thoughts will play a critical role in psychological adaptation to the start of aging. Attitude toward own aging (ATOA) is one example of such potentially positive thoughts. ATOA refers to expectations of the personal experience of aging [36] and it consists of a personal evaluation of age-related changes [3]. Although the ATOA scale has been widely used among older adults [37], recent work also emphasizes its focus on development processes in midlife [3,38,39]. 

We chose ATOA as an intermediate factor to link dance and greater quality of life for two reasons. First, ATOA arises from multiple factors including physical changes, sociodemographic variables [40], and psychological resources [41], and how it develops in mid-adult years is not well-understood yet [3]. Meanwhile, ATOA is proven to be influenced by the feeling of functional actions in even small activities of daily living [42]. As a result, applying positive practices might partially explain how ATOA variation is shaped when the new phase of life is approaching. When people are practicing positive activity, they make relatively more positive appraisals of their life events [5], and their efficacy and positive thoughts will become the foundation of life attitude. By performing Chinese square dance, people experience a positive change in life which is crucial for raising positive expectations of life experience [43]. Since dancers are experiencing a life change toward aging, their choice in practicing positive activity via Chinese square dance could facilitate the generation of positive ATOA.

Secondly, ATOA maintenance is also important when physical functioning declines [41]. People have needs for autonomy [44] and belongingness [45], which are essential for psychological health and facilitate effective functioning in social settings [46]. For aging adults, the desire to maintain autonomy and control is also stronger [47]. At this point, mechanisms of positive activities start to work. When people are intrinsically motivated to pursue enjoyable activity, their positive behaviors give them feelings of autonomy and increase their self-control [48], while the control beliefs are important psychological resources for preserving positive ATOA [49]. Meanwhile, engaging in positive activities could satisfy multiple needs [5], including autonomy (control), competence (efficacy), and relatedness (connectedness).

In regards to Chinese square dance, the connectedness is featured by social support [15], and perceived social support greatly affects ATOA [50]. In summary, engagement in Chinese square dance is essential to generate and maintain positive ATOA. Therefore, Hypothesis 2 assumes that time spent on Chinese square dance is positively related to ATOA. 

Meanwhile, ATOA is a good predictor of life satisfaction, which is the main representation of QoL. Researchers have shown that ideas about own aging have a long effect on physical function and emotional well-being, which is often viewed as the psychological component of QoL [51,52]. To be more specific, ATOA has been proven beneficial to life satisfaction and positive affect [53,54,55]. When people hold a more positive ATOA, they have been observed to have better life experiences. As a result, Hypothesis 3 assumes that ATOA would mediate the positive relationship of Chinese square dance duration to QoL. 

## 4. Moderation by Aging Stereotype: A Buffering Mechanism

Although the duration of dance (a positive and emotionally meaningful activity) is expected to be associated with higher levels of ATOA, whether this association is determined by personal features is yet to be identified. According to the positive activity model, attributes of the person engaging in the activity impact how people benefit from positive practices. For example, people’s motivation to continue [56], personalities [57], and their initial affective state [58] may affect how and when they gain from positive activities. ATOA starts to gain importance in the midlife phase because the experience of a new framing of lifetime emerges, and large interindividual differences in physical changes and age perception might cause large variations in ATOA [4]. In this way, the association between positive activity and ATOA might depend on the degree to which people perceive the aging attitude from their own cognitive stereotypes. 

The aging stereotype refers to people’s inherent notion and expectation of elderly individuals [59], which reflects people’s view of traits associated with old age and expectations about aging. Both positive (e.g., wise) stereotypes and negative (e.g., senile) stereotypes occur to human beings. They have opposite impacts on people’s performance and behavior, and are related to health in contrasting ways [60,61]. Negative age stereotypes are more in prevalent in society [62,63] than positive ones, however, people with different traits [64] could present variations in aging stereotypes. Many people have negative responses regarding their distant future, while some individuals harbor a preponderance of positive aging stereotypes [65] and are willing to maintain and foster the already existing positive perceptions [66]. Additionally, individuals also show differences in the ability to cope with stereotypes [67] or to respond to the intervention of implicit stereotype priming [68]. Because people’s perceptions of their own aging are also significantly different from their own societal view of aging [40], the inherent interindividual differences in age stereotypes become larger [66], resulting in a discrepancy in people’s innately held view of aging and their capacity to cope with the aging stereotypes.

Other than this, interindividual differences of aging stereotypes also exist among middle-aged and older adults [66]. The inconsistency of stereotypes in different age groups occurs mainly because of the social identity theory regarding age as the basic intergroup cue [69,70]. A salient group identity can serve as the label to indicate the feature of members of an out-group, and age is an especially unique social category. However, this social identity cue is flexible with age change since individuals’ definitions of in-group/out-group categories also change according to their aging process. The change of perception in group identity creates some special findings in middle-aged adults: they still view older adults as members of a different age group when establishing traits for age-related stereotypes [71]. As Heckhausen pointed out, middle-aged adults’ perceptions of the elderly may fall between those of young and elderly adults and lean to one side or the other at certain points of view [72]. Among people in midlife, in other words, stereotypes for the elderly are quite different from the perception of their own aging. 

Aging stereotypes differ across people and between age groups, which led us to further explore them as an individual characteristic variable and to wonder whether the impact of square dancing on ATOA and QOL varies depending on the level of these stereotypes. Early research has proposed that aging stereotypes and ATOA have a causal relationship such that positive stereotypes would lead to more positive ATOA and vice versa [2]. Thus, the importance of aging stereotype differences for ATOA is readily apparent. The interaction of activity duration and aging stereotypes in predicting ATOA and QoL remains unclear. Yet with the model of positive activity, duration of dance and stereotype are considered to be largely independent constructs and powerful indicators for ATOA. More time spent on Chinese square dance is expected to be associated with more positive ATOA, while personal difference in perception of aging stereotypes could moderate the strength of such association. The existing interindividual differences in people’s initial stereotypes, to some degree, determines the extent of their psychological resource and, further, the level of their ATOA [73]. People with a more negative aging stereotype (seeing old as fragile) might benefit more from the positive activity effect: dancing activates the inner resources of people to provide positive belief. On the other side, dancers who have more positive aging stereotype will display attenuated positive activity effect. Such people usually initially keep a positive ATOA and do not need extra effort to prove that they are not disabled. Specifically, it is hypothesized that the mediating effect of ATOA between dance duration and QoL is stronger for individuals with higher levels of negative stereotypes (Hypothesis 4) and weaker for individuals with higher levels of positive stereotypes (Hypothesis 5). Figure 1 presents the research model that explains the main hypotheses of the relationships among all the variables in this study. 

## 5. Method

### 5.1. Participants and Procedure

From April 2017 to May 2018, 496 Chinese square dancers were recruited using a non-random convenience sample. This sample scale was determined by the recommended sample size to conduct structural equation models with nonnormal indicators with missing data [74], due to the nature of QoL measurements. Fifty-five people failed to complete the response process or did not provide correct information; thus, they were excluded from the final analysis, which included 441 valid participants. (The age distribution of this sample was: <40 years 14%; 40–60 years, 77%; >60 years, 8%, which is similar to the report of the large-scale national survey in 2017. Therefore, it is believed that we collected a typical sample of Chinese square dancers.) This number (Mage = 48.8 years, *SD* age = 9.57, 84.1% female, range = 18–76) completed the whole survey and received RMB 100 for compensation. 

The dancers were recruited through an acquaintance circle. First, advertisements about dancing survey were put on the university website to recruit school volunteers who had access to the dancers. After that, student investigators were trained to help dancers with the survey. The participants invited by the trained student volunteers agreed to record their lives for one week and were contacted through the social media platform WeChat. Every night before 10:00 p.m., the registered participants received a text message containing the survey link for that day and answered it before the day ended. The survey contained a diary and a questionnaire section. The diary section recorded basic information about the duration of dancing activity every day, while the questionnaire section measured psychological indicators such as ATOA and aging stereotypes at different times. While answering the survey, the dancers were required to report their participation ID for matching. Except for the participation ID, no personal information was recorded, and we guaranteed confidentiality and anonymity to every participant. Informed consent was obtained through the survey according to procedures approved by the Committee for Protecting Human and Animal Subjects at our university.

### 5.2. Measures

Demographics. At the end of data collection, participants reported their age, gender, family income, marital status, and education data. Participants’ height and weight were requested to calculate body mass index (BMI = weight (kg)/[height (m)]^2^). See Appendix A for more demographic characteristics.

QoL measurement. QoL was measured by the WHOQOL-BREF scale [21] on the last day of the week. WHOQOL-BREF comprised four domains: physical health, psychological health, social relationships, and environmental status. Altogether, 26 items were answered on 5-point Likert scale and the points were linearly transformed to a score from 0 to 100 for each domain. A well-established Chinese version of this questionnaire was adopted in this study.

Image of aging (IOA). Aging stereotypes were measured on the third day using the IOA scale. The IOA scale [75] was used to assess aging stereotypes and the image that older people hold of the aging process. A single question instructed participants to assess how much the items matched their image of “thinking of elderly people in general (not including yourself)”. Choices from 0 = “not at all the way I think” to 6 = “exactly the way I think” were made for a series of words with age-related negative concepts (e.g., walks slowly, helpless, sick) and positive concepts (e.g., active, wise, healthy). The overall score for this scale was determined by summing all the items by category and formed the positive and negative aging stereotype scores.

ATOA. ATOA was measured on the fourth day. The attitudes toward own aging subscale from the Philadelphia Geriatric Centre Morale Scale (PGCMS) was used to assess the self-attitude of aging. Items such as “As I get older, things are better than I thought they’d be” and “I am as happy now as when I was younger” were asked to indicate whether participants agreed (score = 1) or disagreed (score = 0) with these statements [76,77]. The total ATOA score ranged from 0 (most negative ATOA) to 7 (most positive ATOA).

Activity duration of Chinese square dance. The details of how people spent their time were recorded in the diary section through the day reconstruction method (DRM) [78]. The DRM asked respondents to complete a structured self-administered questionnaire to reconstruct their memory of the activities performed on the previous day. A participant first reinstated the previous day into a sequence of episodes and then produced a short diary of the events they experienced. They then referred to their diary notes to describe the key features of each event, including when the episode began and ended, and what they did. The period of time was summed to obtain the whole duration (timing) of an activity. In this way, the duration of Chinese square dance was generated. Later, the time counts were coded into a three-level ordinal measure [79] due to wide variation in the level of engagement. The tertiles of the overall time range were used as cutoffs to divide the time spent on Chinese square dance evenly into three groups, and in terms of “low”, “medium”, and “high” engagement to represent the activity duration (see details in Appendix A).

Control variables. Variables such as age, BMI, and gender were chosen as control variables in this study because the literature had demonstrated their relation to physical activity engagement [80], physical activity stereotypes [81], and health-related behaviors [37]. We assumed that these effects would also apply to square dancers who were typically middle-aged. Other demographics such as education were unrelated to the outcome and thus not included in the final analysis.

### 5.3. Analytical Strategies

The overall time of Chinese square dance was first calculated using R (Version 3.6.0) to make the multicategorical variable. Main hypotheses were tested using SPSS 22.0 and MPLUS version 8.3 [74], and *p* < 0.05 was considered statistically significant. To construct the multiple regression model, indicator coding was used for analyses with multicategorical independent variables. With three levels of dancing duration, two dummy variables (Dummy 1 and Dummy 2) were constructed for the medium and high duration group. The low duration group was not explicitly coded and thus functioned as the reference category in the analysis. The parameters in the model pertinent to group differences were quantifications relative to this reference group. 

Since the subdomains of QoL scale did not show differential expectations in patterns, a latent factor with these subdomains was modeled to test the current mediation and moderation hypothesis. Good model fit was assessed based on the comparative fit index (CFI, ≥0.95), the Tucker–Lewis Index (TLI, ≥0.95), the root mean square error of approximation (RMSEA, ≤0.08), and standard root mean square residual (SRMR, ≤0.06). Confidence intervals (95%) were generated by bootstrapping with 5000 resamples and missing values (<10% on all control variables, *n* = 38) were treated with listwise deletion. 

## 6. Results

### 6.1. Descriptive Analyses and Correlations

Means, standard deviations, Cronbach’s alphas, and bivariate correlations for major variable are presented in Table 1. Pearson’s correlations showed that ATOA was positively correlated with all domains of QoL; and endorsement of a negative aging stereotype was negatively correlated with the ATOA and QoL scores, but a positive aging stereotype was positively correlated with the ATOA and QoL scores. 

### 6.2. Testing the Hypothesized Model

Simple mediation: The integrative structural relations model consisted of QoL as the latent variable, and dance duration, ATOA, stereotypes, and other control variables as observed variables. The model results showed a good fit to the data (CFI = 0.987, TLI = 0.980, RMSEA = 0.034, SRMR = 0.028, χ^2^ (403) = 33.95, *p* = 0.07, χ^2^/df = 1.48). Analyses revealed a main effect of increasing dance duration on QoL, such that participants in the medium dance duration condition (Dummy 1: *B* = 3.51, *SE* = 1.48, *p* = 0.02) and high duration condition reported higher QoL perceptions (Dummy 2: *B* = 3.31, *SE* = 1.50, *p* = 0.03). Therefore, significant positive correlations between dance duration and QoL are tested as expected in Hypothesis 1. However, Hypothesis 2 was only partially proved. The effect of Dummy 1 (*B* = 0.13, CI 95% (−0.33, 0.55)) and Dummy 2 (*B* = 0.56, CI 95% (0.17, 0.96)) on ATOA showed that compared to dancers who had low engagement in Chinese square dance, high-duration dancers showed significantly more positive ATOA but medium-duration dancers did not. The results indicated that only when the time of dance was long enough would the increased dance duration lead to ATOA enhancement.

The indirect effect of Dummy 1 on QoL via ATOA equaled *B* = 0.30, CI 95% (−0.81, 1.27) while the indirect effect of Dummy 2 on QoL via ATOA equaled *B* = 1.27, CI 95% (0.36, 2.40). These results indicated the successful relative mediation of the Dummy 2 variable, which supported that ATOA mediated the relationship between dancing durations (high vs. low) and QoL perceptions, supporting part of Hypothesis 3. Additionally, compared to the reference group, the medium-duration group (Dummy 1) did not show much difference in the perceptions of ATOA, thus the relative mediation of Dummy 1 was not supported. Figure 2 displays the latent variable structural equation model with standardized parameter estimates.

Moderated mediation: The latent QoL construct was regressed on dummy variables, ATOA, and control variables while ATOA was regressed on aging stereotypes (IOA) and the interaction between dance duration and aging stereotypes to test a conditional process model (moderated mediation). Negative and positive aging stereotypes were separate constructs in measurement and were predicted to have opposite directions in their interactions; therefore, multiple regression model analyses were conducted and the results were respectively displayed. 

The first model using negative image of aging as moderator showed a good model fit to the data (CFI = 0.977, TLI = 0.974, RMSEA = 0.034, SRMR = 0.034, χ^2^ (403) = 63.91, *p* = 0.03, χ^2^/df = 1.45). As shown in Table 2, analyses revealed a significant dance duration ×negative image of aging interaction on ATOA (for Dummy 1: *B* = 0.61, *SE* = 0.26, *p* < 0.05; for Dummy 2, *B* = 0.46, *SE* = 0.23, *p* = 0.05). As predicted, when people showed high values of negative image of aging (+1 *SD*), both the medium-duration (*B* = 0.78, *SE* = 0.38, *p* = 0.04) and high-duration dancers (*B* = 1.03, *SE* = 0.36, *p* < 0.01) significantly predicted positive ATOA compared to the reference group. Such effect did not emerge when people showed low levels of negative stereotypes (−1 *SD*), indicating the effect of dance engagement on ATOA was moderated by innate negative aging stereotypes. The interaction is shown in Figure 3, which plots the simple slopes of dance duration on ATOA at various values of negative IOA (mean, ±*SD*).

Further, the moderated mediation analysis (see Table 3) revealed significant conditional indirect effects on QoL via ATOA for Dummy 1 condition (*B* = 1.78, *SE* = 0.92, *p* = 0.05, CL 95% (0.035, 3.651)) and for Dummy 2 condition (*B* = 2.34, *SE* = 0.98, *p* = 0.02, CL 95% (0.671, 4.448)) at high values of negative image of aging (+1 *SD*). That is, when the dancers showed highly negative IOA, more time spent on dance resulted in more positive ATOA, which subsequently increased QoL perceptions. Notably, the conditional indirect effects were not significant (95% CL included 0) when people had low levels of negative IOA. These results supported Hypothesis 4.

The model with positive image of aging as moderator showed a relatively poor but acceptable fit to the data (CFI = 0.900, TLI = 0.887, RMSEA = 0.073, SRMR = 0.093, χ^2^ (403) = 137.87, *p* < 0.01, χ^2^/df = 3.13). The analyses revealed a nonsignificant medium dance duration × positive IOA interaction on ATOA (*B* = −0.29, *SE* = 0.25, *p* = 0.25) and a marginal significant high dance duration × positive IOA interaction (*B* = −0.43, *SE* = 0.22, *p* = 0.06). When people were at various values of positive image of aging, Dummy 1 variable could not predict positive ATOA (*p* > 0.05). Yet the Dummy 2 variable predicted ATOA at low level of positive IOA (−1 *SD*, *B* = 0.83, *p* = 0.01), and the prediction was not significant at high level of positive IOA, which was in accordance with the predicting direction. The details of this model are presented in Appendix A.

Further, the moderated mediation analysis only revealed significant conditional indirect effects of Dummy 2 on QoL at −1 *SD* of positive IOA (*B* = 1.87, *SE* = 0.82, *p* = 0.02, CL 95% (0.316, 3.575)), see details in Appendix A. Therefore, the association between dance duration and QoL via ATOA was also moderated by the level of positive image of aging: when people had less positive IOA, increased dance time had more impact on ATOA and QoL. The results partially supported Hypothesis 5, since the moderation did not emerge for the Dummy 1 variable.

## 7. Discussion

In general, the present analysis of a midlife sample in China presents our understanding of the personal correlates of Chinese square dance and relationships with people’s ATOA in the realm of QoL. Current data analysis reveals the impact of dance duration on QoL, as mediated by attitude toward own aging. Based on the positive activity model [5] and studies indicating that physical activities enhance QoL [20,31], we hypothesize that if people choose to participate in certain positive practices, i.e., Chinese square dance in the context of this research, their perception of good life quality will be enhanced. Evidence from socioemotional electivity theory [10] also proved that the socially meaningful property in this activity serves as the attraction to middle-aged people. Structural model analysis shows that duration in dance is positively correlated with QoL. People with an intention to participate in square dance, as long as they spent time on it, experienced better overall feelings of life quality. That is, the dance practitioners had higher quality of life, and dance duration played a central role.

In addition, the hypothesis that dance duration influences the attitude toward aging is fully confirmed. Our analysis shows that the longer the duration of square dance observed, the more positive were attitudes toward a person’s own aging. When the duration of dance was not at a high level, the ATOA did not change accordingly with the dance participation. This result supports the Need Theory [12] and motivation theory [48]. When social need is satisfied, positive thoughts from the activity become the resource for changes in attitude. The more time spent in gaining happiness, the more positive thoughts are accumulated and the more needs are satisfied. With the beginning of a virtuous circle, the belief-shaping process occurs when the duration of activity becomes longer. The finding is consistent with studies that have demonstrated that behaviors, lifestyles, and action routines are positively associated with attitude formation and shaping [82,83]. 

Furthermore, our study found that the duration of dance can influence peoples’ QoL through their attitude toward own aging. The clear positive dose–effect relationship between Chinese square dance and QoL is largely dependent on the nature of positive activity and the inner mechanism for dance duration to predict ATOA. Since ATOA well predicts the subjective part of QoL, and dance duration also has a significant promoting effect on QoL, our hypothesis confirms that dance duration can influence QoL through its impact on the attitude toward aging. Although this hypothesis differs from the traditional understanding of the mechanisms by which physical activity promotes QoL [20], it takes a psychological perspective to explain the possibility of a mind–body interaction; namely, how actions influence attitudes. The analysis of this study reveals that, by engaging in positive daily living activities in midlife, people are able to have more positive attitudes toward their own aging, and transform it to the positive experience of their physical and social environments.

Another notable finding of this study is to identify the role of aging stereotypes on the association between dance duration and ATOA. The endorsement of aging stereotypes is shown to predict life satisfaction among elderly adults through their ATOA [84]. Meanwhile, in studies of internalization of stereotypes and subsequent well-being [73], the endorsement of aging stereotypes [81] is often considered as a more distal variable to impact the formation of ATOA. Apparently, people with a certain level of aging stereotype are prone to generate similar attitudes toward themselves, and this correlation might change the strength of positive activity effect. It is also interesting to find that dance duration had a weaker impact on the variation of ATOA when considering positive aging stereotypes as a preceding variable. When people incline to view aging positively, to actively engage in square dance seems to have less contribution to their own attitude toward aging. This is interesting because the positive activity model originates from the idea to capture the features of naturally happy people. As positive practices are made more prominent when facing negative traits, it can also be made inconspicuous when facing positive traits. The results of opposite impacts of the two stereotypes reveal that less effort in performing positive activities is needed when we want to help people with negative traits, but more effort is needed to help to maintain or promote those with positive traits. As for the Chinese square dancers, although not too old to be affected by aging beliefs, they were still aware of the coming of that life phase. To dance is probably a strategic way for them to fight against innate stereotypes from society and to form positive self-belief. 

This study makes two primary contributions to the understanding of the relationship between Chinese square dance and QoL. First, results from this research re-examine the dose effect of dancing activity on QoL and explain it with a novel perspective. The literature on attitude formation has noticed a behavior–attitude correlation when attitude is strengthened with direct prior experience [85]. Based on direct experience, attitudes are more likely to be easily accessible [86], and active personal involvement induces individuals to think and form their attitudes [87]. Through the positive activity model, the link between Chinese square dance and positive attitude formation can be understood through the positive thought path. Because a positive attitude is a good predictor of subjective satisfaction and emotional well-being [53], we are able to provide a psychological explanation rather than merely basing the benefit of Chinese square dance on only physical changes. 

The second contribution is the finding on the conditional effects of the aging stereotypes. Studies of stereotype threats [88,89] have shown that activation of negative aging stereotypes significantly degrades the well-being of adults and further influences their subjective well-being because inherent stereotypes color beliefs regarding the capability to live in and manage the future. The decrease of subjective well-being reflects the attenuation of the subjective aspect of QoL. Although positive aging stereotypes are also generated in the process [37], negative views seem to have a stronger influence because of their prevalence. Meanwhile, according to theories for communication in aging [71], and intergenerational interaction [70], aging stereotypes are a barrier in the successful aging process. Self-categorizing as old may raise more negative consequences for people’s emotion and efficacy. At the same time, prejudices between in-groups and out-groups will set apart people of different age groups, and create psychosocial distance between people. Such results are a reminder that the difference in individual aging stereotypes has potential to affect aging-related beliefs, to affect communication between different age groups, and to lower the appeal of social needs to people. Results from this research find that positive activity and aging stereotypes interact to influence the formation of aging attitudes. Additionally, the involvement of aging stereotypes defines the boundary of people who would benefit more from positive-activity intervention. 

The implications also point to suggestions for policies and society. Under the influence of positive psychology, the WHO proposed the policy objective of active aging at the Second World Assembly on Ageing in 2002. Active aging is the process of optimizing opportunities for health, participation, and security in order to enhance quality of life as people age, with the core concept of “health, participation and security” and the creation of a favorable social environment to promote the participation of older people in valuable activities [90]. The framework of our study is in line with the two prospects of the objective. First, we believe the construction of an effective communication environment that fosters positive aging experiences and feelings of efficacy is essential in response to the need to create a suitable social environment for active aging. In the context of this article, it is crucial to understand the significance of age stereotypes in interpersonal communication because how we interact with people of different ages reflects the social identity we feel we belong to. We must stop treating the aged as unable people and start treating them with respect, assisting them to develop efficacy, decreasing unneeded negative aging stereotypes and promoting positive stereotypes so as to enable a successful aging process. The issue is not only important for individual health, but rather gives a healthy tone to the whole community and the social culture. 

Secondly, valuable activity participation for older people is needed. A positive psychological state and positive physical activities are both initiatives for a healthy aging life. Although negative aging stereotypes prevail in society [91], participation in Chinese square dance is a means to ameliorate the negativity and achieve a high QoL. Our suggestion in introducing more interventions like square dance echoes the 2020 WHO guidelines [92], in which the phrase “every move counts” was coined to encourage participation in any type of physical activity at any duration. Though increasing physical activity might be a challenge when people become older, adding more attributes such as positive benefits and emotional meaning to activities would provide the inner motivation to “do the thing”. Our findings might inspire the society to provide more opportunities to advocate and support Chinese square dance, such as offering public spaces for dancing, organizing dancing groups, or advocating Chinese square dance. Meanwhile, more activities that are positive and suitable for older people can be added to the interventions to support citizens to integrate more physical activity in their daily lives.

## 8. Limitations and Further Research

This investigation has several limitations. Because of the correlational nature of the data, inferences on causal relationships between predictors, moderators, and perceived QoL are not fully confirmed. Although the theoretical framework clarified the sequence between variables, a more robust design such as a longitudinal study is necessary to test whether using Chinese square dance as an intervention may change views about aging. Second, this study lacks a matched control group to highlight the effectiveness of the proposed dancing strategy. Though the initial aim of our research was to test the positive activity effect caused by Chinese square dance and did not include group differences, a comparable group is needed in future study design to see whether this effect exists in other populations or in other activities. At the same time, effects caused by other activities need to be considered as controlled, especially when square dance is often regarded as similar to many physical activities [93,94]. Future studies in this field may need to take into account potential participants who engage in more than two types of physical activities or compare groups engaging in different physical activities, such as yoga and square dancing. Third, due to sample convenience, the dance participants were mostly urban residents who resided in nearby provinces, raising concerns about the generalizability of the findings. This type of dance is practiced across China: thus, further research should replicate the present findings in settings with more locations and make comparisons that are more even to test gender differences and urban and rural differences. Fourth, there are limited physiological data to validate the positive effects of square dancing on quality of life, as the objectivity component is also an important component of QoL [26]. In future studies, objective measures as accelerometers can be used to record time spent in dance and thus provide additional information about the intensity of it. Potential physical benefit could be the key to attract people to engage in activities, and the physiological benefits might further affect psychological well-being and then QoL. Fifth, this research reveals individual differences only from the perspective of aging stereotypes. Investigations of individual-level factors, such as personality, motives, and emotions, are necessary to more accurately understand the direct and indirect effects of Chinese square dance on individual quality of life. Using the previous experience in dance as a confounder would also be interesting. Past experience often has deep-rooted influence on attitude forming. If we control for the effects of past dance experiences on ATOA and QOL, we might be able to explore whether hedonic adaptation will happen. Sixth, during the process of measuring IOA, some of our instructions were not in accordance with a more proper way to describe clear requirements for creating age-related traits [71], such as saying “do not think about yourself when describing the typical elderly adult”. We understand the expression of excluding people in the items of a scale may have caused guidance that was consistent with the researcher’s intent. In future work, subjects will be guided by more established research paradigms and measured in ways that do not produce induction. Finally, by summarizing data across days, a methodological limitation is that this research did not make full use of the day-level data. The diary method applied in this study may produce richer findings than those of cross-sectional analysis in the literature, and further research might address the question tapping exposure to more micro-level indications for the benefits of Chinese square dance. 

## 9. Conclusions

In summary, the results of this research indicate that aspects of Chinese square dance, along with levels of negative aging stereotype, are associated with QoL for midlife adults. In line with the literature on socioemotional selectivity theory and positive activity models, we highlight the role of Chinese square dance in the regulation and maintenance of active aging attitudes and propose that certain people holding extreme age-related stereotypes may use dancing activity as a strategy to maintain their health and to pursue a better quality of life. 

## Figures and Tables

**Figure 1 ijerph-19-16477-f001:**
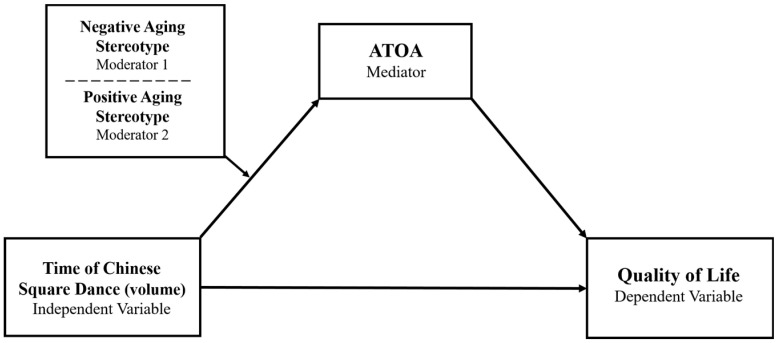
The hypothesized conditional process model when negative and positive aging stereotypes are separately treated as moderators.

**Figure 2 ijerph-19-16477-f002:**
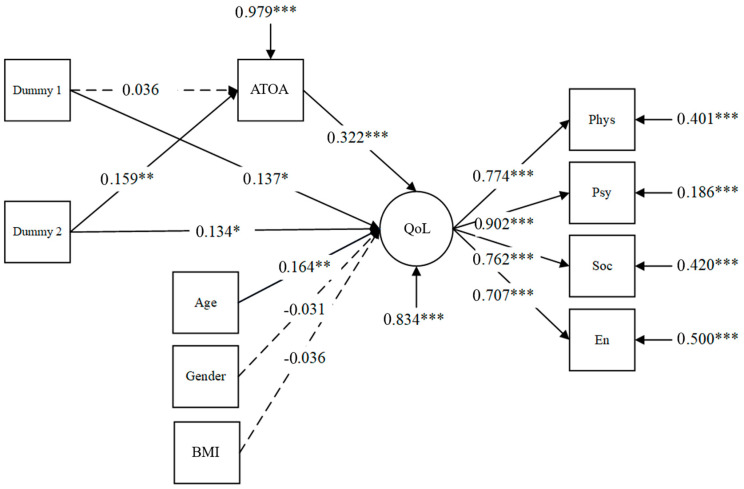
Structural model for QoL. Note: The circles represent latent variables, and rectangles are observed variables. The broken lines indicate effects that are not statistically significant. Path coefficients are standardized MPLUS parameter estimates. The subdomains of QoL scales are: Phys = physical, Psy = psychological, Soc = social, En = environmental. * *p* < 0.05; ** *p* < 0.01; *** *p* < 0.001.

**Figure 3 ijerph-19-16477-f003:**
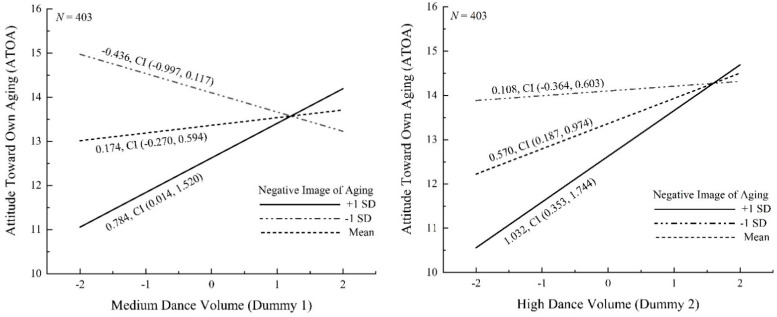
Conditional effects (simple slopes) of dance duration on ATOA at various values of negative IOA (mean, plus, and minus one standard deviation from the mean) for Dummy 1 (**left**) and for Dummy 2 (**right**).

**Table 1 ijerph-19-16477-t001:** Descriptive statistics, alpha coefficients (in parentheses), and correlations.

	M	*SD*	1	2	3	4	5	6	7	8	9	10
1. ATOA	5.32	1.70	(0.661)									
2. IOA-positive	5.52	0.89	0.258 **	(0.847)								
3. IOA-negative	2.92	1.16	−0.222 **	−0.177 **	(0.848)							
4. Qol-physical	70.68	15.65	0.350 **	0.392 **	−0.162 **	(0.776)						
5. Qol-psychological	76.04	15.63	0.300 **	0.462 **	−0.128 **	0.707 **	(0.802)					
6. Qol-social	75.92	15.21	0.241 **	0.377 **	−0.144 **	0.577 **	0.700 **	(0.720)				
7. Qol-environmental	40.94	4.44	0.271 **	0.328 **	−0.125 **	.566 **	0.667 **	0.589 **	(0.810)			
8. Age	48.8	9.57	−0.013	0.004	0.059	0.135 **	0.175 **	0.119 *	0.078	-		
9. Gender	1.84	0.37	0.034	−0.015	−0.035	−0.043	0.011	0.068	0.007	−0.049	-	
10. BMI	23.3	2.93	−0.066	−0.033	0.066	−0.002	−0.024	0.018	0.007	0.197 **	−0.150 **	-

Note: N = 403. The alpha coefficients for the constructs are given in parentheses on the diagonal. For gender, 1 = male, 2 = female. ATOA = attitude toward own aging. IOA = image of aging. * *p* < 0.05. ** *p* < 0.01.

**Table 2 ijerph-19-16477-t002:** Regression results on ATOA in model with negative image of aging.

	*B*	*SE*	*t*	*p*
Predictor				
Constant	12.68	0.99	12.83	<0.001
Age (covariate)	<0.001	0.01	−0.03	0.98
Gender (covariate)	0.04	0.23	0.16	0.88
BMI (covariate)	−0.03	0.03	−0.85	0.40
Negative image of aging	−0.74	0.19	−3.86	<0.001
Medium dance duration (Dummy 1)	0.17	0.22	0.79	0.43
High dance duration (Dummy 2)	0.57	0.20	2.83	0.01
Dummy 1 × negative image of aging	0.61	0.26	2.39	0.02
Dummy 2 × negative image of aging	0.46	0.23	2.00	0.05
Conditional effect of medium dance duration (D1) on ATOA at values of negative image of aging				
Negative image of aging				
−1 *SD*	−0.436	0.288	−1.515	0.130
+1 *SD*	0.784	0.382	2.054	0.040
Conditional effect of high dance duration (D2) on ATOA at values of negative image of aging				
Negative image of aging				
−1 *SD*	0.108	0.247	0.436	0.663
+1 *SD*	1.032	0.356	2.898	0.004

Note. Unstandardized regression coefficients are reported.

**Table 3 ijerph-19-16477-t003:** Regression results on QoL in model with negative image of aging.

	*B*	*SE*	*t*	*p*
Predictor				
Age (covariate)	0.20	0.06	3.29	<0.001
Gender (covariate)	−1.01	1.41	−0.72	0.47
BMI (covariate)	−0.15	0.22	−0.69	0.49
Medium dance duration (Dummy 1)	3.51	1.48	2.38	0.02
High dance duration (Dummy 2)	3.31	1.50	2.21	0.03
ATOA	2.26	0.46	4.96	<0.001
Conditional indirect effect of medium dance duration (D1) on QoL via ATOA at values of negative image of aging				
Negative image of aging				
−1 *SD*	−0.988	0.716	−1.380	0.168
+1 *SD*	1.776	0.919	1.932	0.053
Conditional indirect effect of medium dance duration (D2) on QoL via ATOA at values of negative image of aging				
Negative image of aging				
−1 *SD*	0.244	0.558	0.437	0.662
+1 *SD*	2.338	0.980	2.385	0.017

Note. Unstandardized regression coefficients are reported.

## Data Availability

The data that support the findings of this study are available from the corresponding author, Honghao Zhang, upon reasonable request.

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
