# Peer review of "Dance to Prosper: Benefits of Chinese Square Dance in QOL and the Moderating Roles of Aging Stereotypes"

_ijerph, 2022, doi:10.3390/ijerph192416477_

Round 1
Reviewer 1 Report
Dear Authors,
Neat study! Here are my suggestions....
Arguments/Lit reviews and discussion about the literature:
p. 2 Given that multitasking, as commonly defined, is really not optimal, and some consider it impossible (i.e., Manhard, 2004), we might consider another term, or just rely on your description of employing multiple physical, mental, and emotional components at once (not actually enacting distinct tasks at once).
Much of what you referene is in accordance with Cartsensen’s work – the positivity bias, the enhanced meaningfulness. It may be useful to tuck alongside works by Lyobomirsky and others on page 2.
When talking about belongingness –in the literature review and/or in the discussion section – you might be wellsuited to reference social identity theory (Tajfel & Turner, 1986) and or the plethora of research on age as a social identity (Harwood, Giles, Hummert are all great scholars to draw from). Hummert is a MJUST for talking about age stereotypes, for she is the founder of their study. Her illuminated stereotypes particular to age would be VERY useful in your literature review and discussion section. She also has a model that employs these that would be worth using your discussion section.
Method:
This “not thinking of yourself” is a strange addition to how we usually study stereotypes. The BIAS map and Hummert’s work both give great eamples of how to reduce social desirability without directly asking them to remove themselves from the equation. If you aren’t going to change this, which I understand because that would involve more data collection, then you should talk about this in the limitations section.
It's great you collected a bunch of data, might their surrounding/other types of exercise be worth addressing, controlling for? I THINK you did b/c you mention “physical activity behaviors” but this isn’t crystal clear and it needs to be so that someone could replicate it.
Results and discussion:
The role of aging stereotypes is meaningful here and the “communication/social psych of aging” audience is in need of a deeper link to the aforementioned models to give these neat findings a bitmore context. You might also draw from activity theory.
I know you have very limited space but please consider these as you move forward so that your important piece can best contribute within and through what else we know! I have given some article suggestions below that I hope might be helpful.
Also, return to the notion even more strongly that this isn’t just healthy for individuals, it is healthy for older adults and for this cultural! ALL benefit from the dancing at enforces identity which is not just an individual thing, also a group thing
References
Manhartt, K. (2004). The Limits of Multitasking. Scientific American Mind, 14(5), 62–67. http://www.jstor.org/stable/24997557
Suggested readings/articles to consider including:
References
Barker, V., & Giles, H. (2003). Integrating the Communicative Predicament and Enhancement of Aging Models: The Case of Older Native Americans. Health Communication, 15(3), 255-275. https://doi.org/10.1207/S15327027HC1503_1
Carstensen, L. L. (1992). Social and emotional patterns in adulthood: Support for socioemotional selectivity theory. Psychology and Aging, 7, 331-338.
Carstensen, L. L., & Hershfield, H. E. (2021). Beyond Stereotypes: Using Socioemotional Selectivity Theory to Improve Messaging to Older Adults. Current Directions in Psychological Science, 30(4), 327-334. https://doi.org/10.1177/09637214211011468
Fowler, C., Giles, H., & Gasiorek, J. The role of communication in aging well: Introducing the Communicative Ecology Model of Successful Aging. Communication Monographs.
Harwood, J., Giles, H., & Ryan, E. B. (1995). Aging, communicaton, and intergroup theory: Social identity and intergenerational communication. In J. F. Nussbaum & J. Coupland (Eds.), Handbook of communication and aging research (pp. 517). Lawrence Erlbaum Associates.
Havighurst, R. J. (1961). Successful aging. The Gerontologist, 1, 8-13.
Hummert, M. L. (1990). Multiple stereotypes of elderly and young adults: a comparisonof structure and evaluations. Psychology and Aging, 5(2), 182-193.
Hummert, M. L. (1994). Stereotypes of the elderly and patronizing speech style. In M. L. Hummert, J. M. Wiemann, & J. F. Nussbaum (Eds.), Interpersonal communication in older adulthood: Interdisciplinary theory and research (pp. 162-185). Sage.
Hummert, M. L., Garstka, T. A., Ryan, E. B., & Bonnesen, J. L. (2004). The role of age stereotypes in interpersonal communication. In Perspectives on Intergenerational Communication (pp. 91-114).
Hummert, M. L., Garstka, T. A., & Shaner, J. L. (1997). Stereotyping of older adults: The role of target facial cues and perceiver characteristics. Psychology of Aging, 12, 107-114. https://doi.org/0882-7974/97/S3.00
Hummert, M. L., Garstka, T. A., Shaner, J. L., & Strahm, S. (1994). Stereotypes of the elderly held by young, middle-aged, and elderly adults. Journal of Gerontology, 49, P240-P249.
Hummert, M. L., Nussbaum, J. F., & Wiemann, J. M. (1994). Interpersonal communication and older adulthood: An introduction. In M. L. Hummert, J. M. Wiemann, & J. F. Nussbaum (Eds.), Interpersonal Communication in Older Adulthood (pp. 1-14). Sage.
Hummert, M. L., Wiemann, J. M., & Nussbaum, J. F. (1994). Interpersonal communication and older adulthood. Sage.
Ryan, E. B., Giles, H., Bartolucci, G., & Henwood, K. (1986). Psycholinguistic and social psychological components of communication by and with the elderly. Language and Communication, 6, 1-24. https://doi.org/10.1016/0271-5309(86)90002-9
Reviewer 2 Report
Review : Dance to prosper : Benefits of Chinese square dance on QOL and the moderating roles of Aging stereotypes.
The first read through from the initial impression seems to be consistent and thorough. It is well constructed and easy to get through and follow along each section. From the abstract, it is relevant and interestingly put to introduce the relationship between dance duration, ATOA, aging stereotypes and QOL. Overall, the paper is well written and comprehendible.
In my opinion, there are successful aspects in terms of information gathered such as sufficient data, and well-constructed table. For example, the ones presented on page 7, table 1 and page 10, table 2. The descriptive statistics and regression results on ATOA are well defined.
Concerning the aspect of English, it is impressive and assured in terms of grammar and vocabulary. Each sentence is well constructed. Conclusion is brief, well summarized and concise.
To sum up, this article should be accepted to be published as it is an outstanding, skilled write up with superlative contents.
Author Response
Response: Thank you for the comments.
We genuinely appreciate you praising and acknowledging our efforts. We really appreciate your feedback on the article's outline, writing style, and research approach.
Additionally, we enhanced the language of the article, added to and expanded the article's literature review and discussion, and considered the thorough feedbacks from other reviewers. All of these improvements have made the article better.
Thank you again for your comprehensive and detailed review work!
Round 2
Reviewer 3 Report
The authors have done a great job reviewing the manuscript. The quality of the manuscript has improved considerably and all of the suggestions and errors that I had previously have been solved.
Only in line 254 they must change "dairy" by "diary".
To my mind the manuscript is ready to publish.